# Regionally sourced bioaerosols drive high-temperature ice nucleating particles in the Arctic

Gabriel Pereira Freitas [1,2], Kouji Adachi [3], Franz Conen [4], Dominic Heslin-Rees[1,2], Radovan Krejci [1,2], Yutaka Tobo [5,6], Karl Espen Yttri[7] & Paul Zieger [1,2] ✉

Primary biological aerosol particles (PBAP) play an important role in the climate system, facilitating the formation of ice within clouds, consequently PBAP may be important in understanding the rapidly changing Arctic. Within this work, we use single-particle fluorescence spectroscopy to identify and quantify PBAP at an Arctic mountain site, with transmission electronic microscopy analysis supporting the presence of PBAP. We find that PBAP concentrations range between $10^{-3}$–$10^{-1}$ $L^{-1}$ and peak in summer. Evidences suggest that the terrestrial Arctic biosphere is an important regional source of PBAP, given the high correlation to air temperature, surface albedo, surface vegetation and PBAP tracers. PBAP clearly correlate with high-temperature ice nucleating particles (INP) (>-15 °C), of which a high a fraction (>90%) are proteinaceous in summer, implying biological origin. These findings will contribute to an improved understanding of sources and characteristics of Arctic PBAP and their links to INP.

Aerosol particles play an important role in cloud formation by acting as cloud condensation nuclei or ice nucleating particles (INP)[1,2]. In addition to suitable meteorological conditions, the formation of cloud droplets and ice crystals is determined by the physiochemical properties of the available aerosol particles, such as their size, concentration, morphology and composition[3,4]. Aerosol particles and their interactions with clouds are considered to be one of the main sources of uncertainty in future climate model predictions[2]. In remote regions with lower aerosol concentrations, such as the Arctic, even subtle changes in aerosol particle sources can have significant impacts on cloud properties, such as cloud phase, radiative properties, cloud lifetime, and precipitation[5,6]; These, in turn, are key elements in the Arctic amplification phenomenon[7,8]. With the current accelerated warming trends in the Arctic, the occurrence of open oceans[9], leads[10], greener tundra[11] and snow-free land is becoming more common

throughout the Arctic[12]. With these changes in mind, certain sources of aerosol may become more prominent, e.g., sea spray[13,14] and/or mineral dust[8,15,16]. Primary biological aerosol particles (PBAP) such as bacteria, spores, pollen, plant debris, or viruses[17] are emitted from vegetation and biological activity on snow and barren land alongside co-emission with sea spray and dust[18,19]. PBAP are poorly understood, given the challenges in sampling and quantifying their presence, emissions and sources[20], especially in the Arctic, where observations are scarce and concentrations are deemed low[21,22]. PBAP found in the Arctic can be of both marine (e.g., sea spray and sea ice) and terrestrial origin (e.g., soil and tundra vegetation), locally sourced or transported to the Arctic from lower latitudes[21].

Previous studies have reported the presence of PBAP in the Arctic[19,21,23–26], together with qualitative assessments of their taxonomy[22,26]. However, these studies were limited in quantifying

[1]Department of Environmental Science, Stockholm University, Stockholm, Sweden. [2]Bolin Centre for Climate Research, Stockolm University, Stockholm, Sweden. [3]Department of Atmosphere, Ocean, and Earth System Modeling Research, Meteorological Research Institute, Tsukuba, Japan. [4]Department of Environmental Sciences, University of Basel, Basel, Switzerland. [5]National Institute of Polar Research, Tachikawa, Japan. [6]Graduate University for Advanced Studies, SOKENDAI, Tachikawa, Japan. [7]The Climate and Environmental Research Institute NILU, Kjeller, Norway. ✉e-mail: paul.zieger@aces.su.se

PBAP sources due to the limitation with the filter sampling methods used[20]. An earlier study has used spore traps to quantify spores and pollens in the Arctic, demonstrating their dominant presence in summer when the local flora is active[25].

Bacteria and, to some extent, fungal spores are known to be efficient INP[21,27,28]. This is due to their size, morphology and composition and especially their excretion of ice nucleating macromolecules[29,30]. Although PBAP are generally expected to be present in low concentrations, they could still be impactful, as INP at remote sites are known to play an important role in determining the cloud phase[31]. Recent studies have linked high-temperature INP (activation temperature > −15 °C) in the Arctic to terrestrial and oceanic biological emissions, but were unable to quantitatively evaluate their sources[21,32,33].

In this study, we used a single-particle fluorescence spectroscopy instrument (multiparameter bioaerosol spectrometer, MBS)[34] to identify, quantify and differentiate PBAP at an Arctic mountain site over the course of one year covering all seasons. PBAP identification was confirmed by microscopy and PBAP tracers. A clear seasonality of PBAP concentration was observed, peaking in summer. The onset of increased PBAP presence followed the depletion in surface albedo due to the retreat in snow cover and correlated with air temperature and vegetation index. PBAP concentrations showed an inverse dependence to wind speed and an anti-correlation with sea salt aerosol tracers. We attributed therefore most of the detected PBAP in summer to originate from regional terrestrial sources. Two methods were used to retrieve INP concentrations at the site. Here, we present strong evidence that PBAP were the main contributor to the concentration of INP active at higher air temperatures. These results will improve the source attribution of PBAP and INP in the Arctic and help constrain their respective representation in climate models.

## Results and discussion
### Seasonal cycles of fluorescent and ice nucleating particles in the Arctic

At an Arctic mountain site in Svalbard, the mean concentration of coarse particles (CP; optical diameter $15 > D > 0.8\ \mu m$) during the year was $330\ L^{-1}$ (interquartile 'IQ' range: $80–343\ L^{-1}$) which correlated to particulate sodium ($Na^+$, Spearman coefficient ($R$) = 0.76, Fig. S1a in the supporting information, SI) and wind speed (Fig. S2a). Hence, most of the CP are believed to have been derived from sea spray[35]. These observations are supported by a 5-day back trajectory source analysis, which revealed that most air masses traveled over ice in winter and over the ocean in summer and fall (Fig. S3a). CP were generally more abundant in winter than in summer. The concentration of fluorescent particles (FP) followed the same seasonal cycle as the concentration of CP (Figs. S1b and S2b), with an annual mean of $26\ L^{-1}$ (IQ range: $7–28\ L^{-1}$). In general, the FP concentration contributed approximately 8% to the CP concentration and this relative contribution remained almost constant throughout the year. A similar pattern of constant contribution of faintly FP has been seen in Antarctica[36], the Pacific Ocean[37], the Southern Ocean[38] and the Baltic Sea[39]. This pattern suggests that over or near the oceans, a certain percentage of particles will always show weak fluorescence, perhaps due to organics present in sea spray aerosol[40,41]. The relative contribution of FP to CP increased only during long-range transport events in summer and fall which were characterized by elevated concentrations of the biomass burning tracer levoglucosan[42] (Fig. S1c). These events were associated with terrestrial sources (Fig. S3a) and originated from northern Europe and Russia with a significant contribution of forest fires[43] (Figs. S4–S6).

FP with high fluorescence (see Methods) were subdivided into highly fluorescent particles (HFP) and fluorescent PBAP. Their differentiation was based on the center wavelength of fluorescence. Those with the main fluorescence signal at 364 nm were classified as PBAP, and all others as HFP. This was based on previous work by ref. 39. The

concentration of HFP (mean: $0.2\ L^{-1}$, IQ range: $0.02–0.1\ L^{-1}$, Fig. 1A) showed distinct peaks with values up to $1\ L^{-1}$ occurring mainly in summer. Elevated HFP numbers were closely related to long-range transport events and elevated levoglucosan concentrations (Fig. 1A). At the beginning of the year, the concentrations of equivalent black carbon (eBC, Fig. 1A) were elevated due to the Arctic Haze phenomenon, which describes the accumulation of anthropogenic aerosol from long-range transport in the first half of the year[44]. The levoglucosan concentrations were as well elevated in January and February (see Fig. S3a) most likely due to transport of biomass burning aerosol from the continents[45]. The HFP concentration for this period followed the eBC concentration ($R$ = 0.84 for January–June) and stayed at around $10^{-1}\ L^{-1}$. For the second half of the year (July–December), the HFP concentration increased to values as high as $10^{1}\ L^{-1}$ and continued to show a high correlation to eBC ($R$ = 0.69). The concentration of HFP did not show any clear dependence on wind speed in contrast to the concentrations of CP and FP (Fig. S2c). In fact, HFP concentration were higher for lower wind speeds, suggesting that HFP would not have sea spray or mineral dust as primary sources. This might indicate that HFP are either locally produced by non-wind driven mechanisms or long-range transported. During the second half of the year, the elevated concentrations of HFP coincided with elevated levoglucosan concentration (Fig. 1A). The long-range transport of biomass burning aerosol[46] could have carried primary particles containing polyaromatic carbon chains that possess strong fluorescence properties[47]. These findings could indicate that the MBS is capable of differentiating Arctic haze and biomass burning particles from PBAP. This is promising, as interference from non-biological FP is a problem often associated with the identification of PBAP using fluorescence-based measurements[40,48].

PBAP concentration (mean: $0.02\ L^{-1}$, IQ range: $6 \cdot 10^{-3}–3.5 \cdot 10^{-2}\ L^{-1}$, see Fig. 1B) had a clear seasonal cycle, with a minimum in winter and spring with levels occasionally $> 10^{-2}\ L^{-1}$ and with a maximum in summer ($10^{-1}\ L^{-1}$). The MBS only detects fluorescent PBAP, which implies that the PBAP concentrations presented here were inherently underestimated[49]. The PBAP concentration numbers ($10^{-2}–10^{-1}\ L^{-1}$) were quite close to those found by Johansen et al.[25] using spore traps in Ny-Ålesund. In addition, the MBS-retrieved PBAP classification and concentration were corroborated by transmission electronic microscopy (TEM) imaging. Arctic PBAP sampled on the 7th August 2020 (red star in Fig. 1B) are shown in panels E–H of Fig. 1. The link between MBS and TEM derived PBAP is described in Section S1 of the SI. The increase in PBAP concentration in June coincided with an increase in ambient air temperature. The same pattern is observed when comparing PBAP emissions with an increase in satellite-derived vegetation index pixel density (e.g., on a scale from 0, no green pixels around Ny-Ålesund, to 1, the maximum number of green pixels for the year 2020). The onset of increased PBAP levels also coincided with a decrease in local surface albedo, snow melt and the decrease in snow depth observed in 2020[50]. Figure S2d clearly shows that PBAP concentration did not depend on wind speed, suggesting that they were not mainly co-emitted with wind-driven aerosol. These findings could indicate a significant influence of local terrestrial biota-driven emissions[51].

To assess sources, the concentrations of the main particle groups (CP, FP, HFP and PBAP) were analyzed with respect to air mass origin on a normalized scale, where negative fractions signified a terrestrial source and positive fractions an oceanic source. Figure S3b shows that PBAP were only slightly connected to the back trajectories of more terrestrial regions. However, since the difference in concentration of CP, FP and HFP segregated by the terrestrial and oceanic contribution is not sufficiently large, reasonable conclusions on the sources cannot be drawn for these 3 groups of particles from the trajectory analysis. HFP on the other hand, were clearly connected to terrestrial-dominated trajectories, possessing concentrations at least one order of magnitude higher than oceanic trajectories.

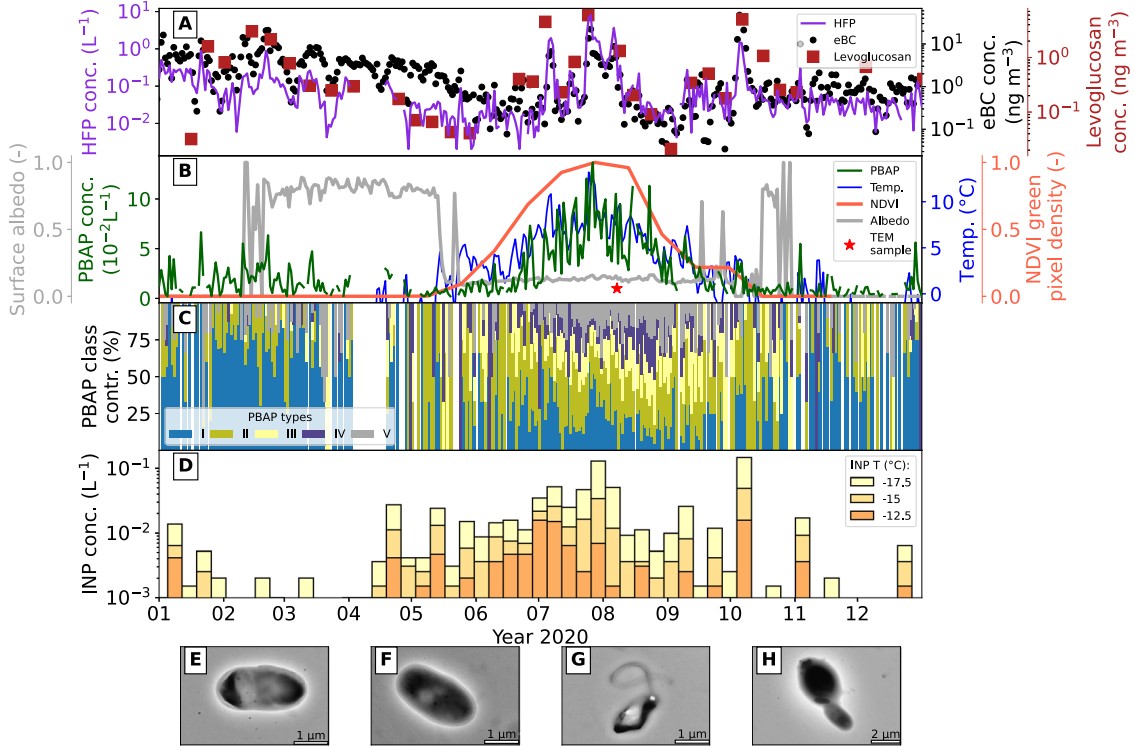

**Fig. 1 | Timelines of several physical, aerosol and INP parameters measured during the year 2020 of the NASCENT campaign. A** Highly fluorescent particle (HFP) concentration, equivalent black carbon (eBC) and levoglucosan concentration. **B** Primary biological aerosol particle (PBAP) concentration, normalized difference vegetation index (NDVI) green pixel density, surface albedo and temperature at Ny-Ålesund. Red star indicates the collection of transmission electron microscopy (TEM) grids. **C** Spectral classes (PBAP types) contributing to overall PBAP signal. **D** Ice nucleating particle (INP) concentration at different nucleating temperatures. **E–H** show TEM images of PBAP sampled on the 7th of August, 2020. The remaining PBAP images for this day are shown in Fig. S12. Source data are provided as Source Data file.

The differentiation of PBAP into several spectral types based on their fluorescence spectra (I–V, Fig. S7) is shown in Fig. 1C. It revealed that each respective spectral type experienced its own clear seasonal cycle. In spring and winter, type I dominated, while in summer types II–V prevailed. Types IV and V were significantly present only in summer. The seasonal cycle observed for the different spectral types of PBAP may have reflected a shift in sources throughout the year, with oceanic and snow sources being predominant in winter and both local terrestrial, oceanic, and long-range transport emissions present in summer, when the PBAP spectral population was more diverse.

Weekly concentrations of high-temperature INP (active above −15 °C, using Method I) classified by activation temperature are shown in Fig. 1D together with those active at −12.5 °C and −17.5 °C. High-temperature INP were much more abundant in summer than in winter, which was also observed by ref. 52. The high-temperature INP concentrations were between $10^{-3}\,L^{-1}$ and $10^{-1}\,L^{-1}$, and had a clear onset and decline that followed the air temperature and surface albedo. In addition, the high-temperature INP concentrations were similar in number and seasonal variation as PBAP. For weeks where the HFP concentration was high, the INP showed clear spikes (e.g., last week of July and first week of October, 2020), indicating the influence of long-range transport of biomass burning aerosol.

**Multi-annual analysis of proteinaceous INP fraction and PBAP tracers**

Over 4 years (2017–2020), we measured the concentration of INP (utilizing INP Method II, at activation temperature −12 °C), as well as their proteinaceous (heat-labile) fraction (Fig. 2A). There was a clear increase in all summers for INP concentrations and their proteinaceous fraction, as seen for PBAP (Fig. 2B). The proteinaceous fraction reached values greater than 90% in summer and ranged between 50% and 85% even in winter. This implies that even in months with little biological activity, most high-temperature INPs were still driven by biological sources, in agreement with previous studies[33,53].

Monthly mean INP concentrations ($10^{-3}\,L^{-1}$, at −12 °C) were of the same order of magnitude as the PBAP concentration and followed the same seasonal cycle. It is likely that PBAP dominated the contribution to high-temperature INP, even in winter. Sze et al. observed a similar seasonal cycle of high-temperature INP for northern Greenland[53]. For three years, the proteinaceous fraction of high-temperature INP was always above 50% and peaked in summer[53]. The same seasonal cycle was also observed near the ice edge[33]. This implies that INP concentrations are strongly influenced by PBAP for a larger part of the Arctic.

The aggregated monthly mean concentrations of arabitol and mannitol for 2017–2020 are shown in Fig. 2C. These sugar-alcohols are tracers of fungal spores[54] and had a similar seasonal cycle to PBAP concentration, high-temperature INP concentration, INP proteinaceous fraction and air temperature. Combining online measurement of PBAP with indirect measurements via PBAP tracers is done in the Arctic for the first time here, tracing the biological source of high-temperature INP to local terrestrial sources. Oceanic sources seem to play a less significant role than terrestrial sources despite the fact that the ocean around the Svalbard archipelago is ice-free for most of the year[55]. High-temperature INP have been attributed to marine biological activity in the Arctic[33], using the oceanic concentration of chlorophyll as a biological tracer. However, elevated high-temperature INP concentrations were observed near a terrestrial source (Svalbard archipelago), and chlorophyll has been shown to be an unreliable proxy for PBAP and organic sea spray emissions[39,56,57] and often requires variable time-lag adjustment[58,59]. Furthermore, over the Pacific and Atlantic

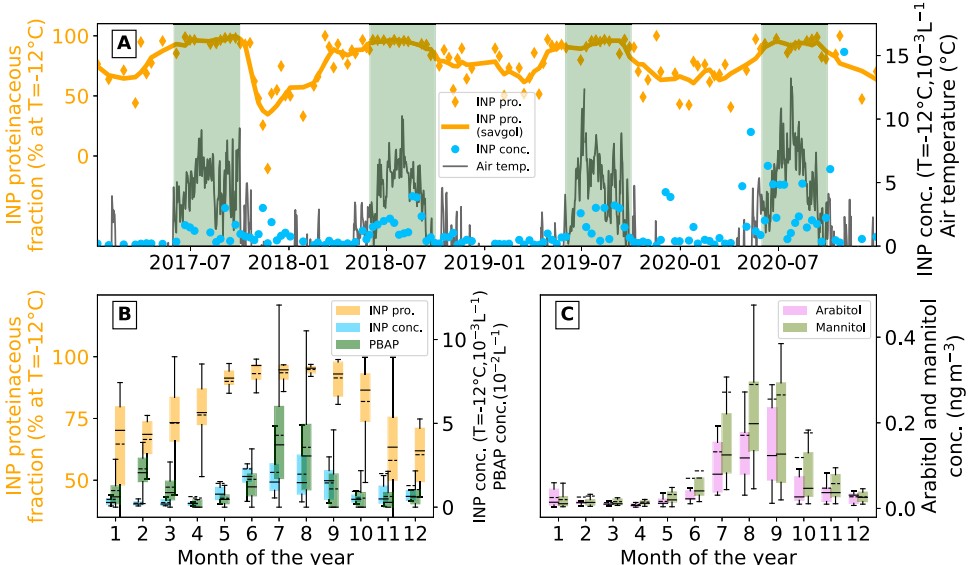

**Fig. 2 | Multi-year trends of ice nucleating particles (INP) proteinaceous fraction and primary biological aerosol particles (PBAP) tracers. A** Timeline of INP proteinaceous fraction (at $T = -12\,°C$) along with a Savitzky-Golay filter curve (savgol) of third order. Green background represents summer months (1st of June until the end of September) with elevated PBAP concentration as it was observed in the year 2020 by the MBS; Along with the observed air temperature (above $0\,°C$) at Ny-Ålesund and INP concentration (at $T = -12\,°C$). **B** Monthly trend of INP proteinaceous fraction and concentration (at $T = -12\,°C$) for four years (2017–2020) along with the PBAP concentration (2020 and some months from 2019 to 2021). **C** Monthly concentrations of arabitol and mannitol measured over the same 4 years as panel (**A**) and (**B**). For the boxplots, the continuous line represents the median and the dashed line the mean, the extent of the colored box shows the interquartile range and the whiskers the data range (1.5 times the interquartile range from the nearest quartile). Source data are provided as Source Data file.

oceans, the airborne microbial populations were found to be strongly influenced by terrestrial sources[60].

## Comparison of particle subsets to INP and other relevant variables

A correlation analysis with other independent measurements was performed to further explore the properties and origin of the MBS-classified particles. The MBS data was aggregated to match the temporal resolution of the other parameters, and we determined the Spearman correlation coefficient and significance ($p$-value)[61]. The particle subsets used were the aforementioned CP, FP, HFP, PBAP and PBAP sub-types. The analysis was extended to include highly asymmetrical CP (HA), elongated CP (EL), and CP larger than 5 and 10 µm. The last four groups were used as a proxy for mineral dust, which is known to be represented by irregularly shaped and/or large particles[62].

The concentrations of CP and INP were not positively correlated at any nucleation temperature (Fig. 3A, utilizing INP Method I). The MBS sampled behind a whole air inlet, while INP filters were collected behind a $PM_{10}$ inlet (see Methods). Nonetheless, 99.997% of the CP were sized by the MBS between 0.8 and 10 µm and thus almost all particles would be sampled by both inlets. The correlation between CP and the sea salt constituents $Na^+$ and $Mg^{+2}$ ($R = 0.76$ and $R = 0.70$, respectively) strengthen the connection to sea spray. A correlation ($R = 0.63$) was found for CP and eBC, indicating either a contribution of Arctic haze to CP or co-location in time of the Arctic haze phenomenon and strong storms during winter and early spring. For subsets of CP (HA, EL, > 5 µm and > 10 µm), the correlations with lower-temperature INP (activation temperature $< = -20\,°C$) increase. However, correlations with sea spray tracers weaken, while correlations with just $Ca^{2+}$ increases, indicating that large and irregularly shaped particles are possibly mineral dust. The latter has been connected to INP in Svalbard[63], and this link appears to be reflected here.

The comparison of some mineral dust tracers with high-temperature INP is shown in Fig. S8. Some spikes in INP concentration are concurrent with spikes in mineral dust tracers, but no matching seasonality is observed. Unlike INP, mineral dust has been shown to have a minimum in summer[64]. Sporadic events with high INP activity could be explained by mineral dust emissions[63] or long-range transport[65, 66]. However, they cannot replicate the seasonality of the high-temperature INP observed here and in other studies[33].

FP correlated with eBC, sodium and magnesium as seen for CP and had a low correlation with levoglucosan, but not with any other organic tracers or INP concentrations at different temperatures, indicating that FP are indeed a subset of sea spray particles and whose contribution to CP is exacerbated during long-range transport events of biomass burning. HFP showed a low correlation with some PBAP tracers such as arabitol and fructose along with organic carbon (OC). HFP presented a high correlation with levoglucosan ($R = 0.63$) and eBC ($R = 0.69$). Like levoglucosan, some PBAP tracers (particularly fructose) are also elevated in biomass burning aerosol[67]. HFP showed some correlation with INP ($R = 0.50$ for $T = -25\,°C$), reflecting the influence of long-range transport events on INP concentrations in the Arctic.

PBAP type I was moderately correlated with fructose and levoglucosan ($0.4 < R < 0.6$) and only weakly ($R < 0.4$) to other organic and PBAP tracers. Type I was present throughout the year and could have also have been brought to the site by long-range transport. In addition, misclassified organic particles with similar fluorescence present in biomass burning aerosol could have contributed to these correlations. For type I, no significant correlations with INP concentrations were found.

PBAP type II correlated with PBAP tracers (mannitol, fructose and arabitol; $R = 0.80$, $R = 0.76$ and $R = 0.70$) and biological secondary organic aerosol tracers (2-methylerythritol, $R = 0.77$). Only low correlation was observed with the proteinaceous fraction of high-temperature INP (INP method II). The same link with PBAP tracers was found for types III and IV, along with a high correlation with the INP proteinaceous fraction ($R = 0.73$ and $R = 0.75$, respectively). Lastly, PBAP type V showed high correlations with PBAP tracers but no significant correlation with the proteinaceous fraction of INP.

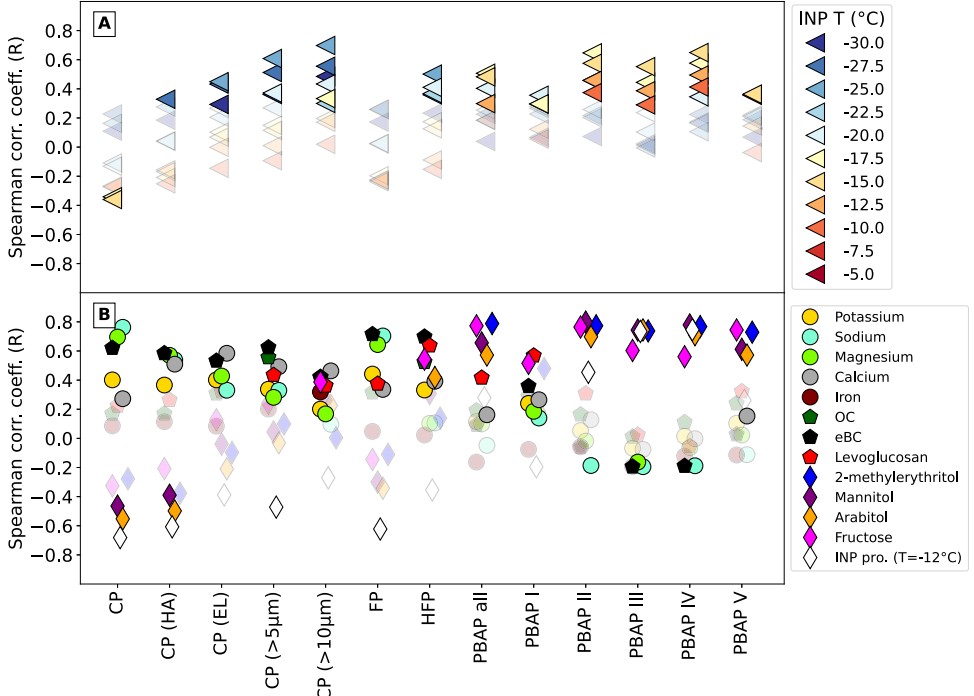

**Fig. 3 | Correlation coefficients between ice nucleating particles (INP), particulate matter ionic/molecular concentrations and the multiparameter bioaerosol spectrometer (MBS) particle groups and spectral classes.** **A** Spearman correlation coefficient (R) between INP concentration at temperature bins and the particles classified by the MBS. **B** Spearman correlation coefficient (R) between relevant ionic/molecular concentrations in particulate matter. the different MBS particle groups and spectral classes are: coarse particles (CP), highly asymmetrical coarse particles (CP HA), elongated coarse particles (CP EL), coarse particles larger than 5 μm (CP > 5 μm), coarse particles larger than 10 μm (CP > 10 μm), fluorescent particles (FP), highly fluorescent particles (HFP), all primary biological aerosol particles (PBAP all), primary biological aerosol particles of spectral class I (PBAP I), primary biological aerosol particles of spectral class II (PBAP II), primary biological aerosol particles of spectral class III (PBAP III), primary biological aerosol particles of spectral class IV (PBAP IV) and primary biological aerosol particles of spectral class V (PBAP V). Solid color markers represent significant correlation ($p < 0.05$). Circle markers represent non-specific tracers. Pentagon markers represent tracers linked to biomass burning and/or anthropogenic aerosols including organic carbon (OC) and equivalent black carbon (eBC). Diamond markers represent primary biological or organic tracers and ice nucleating particles (INP) proteinaceous fraction (INP pro.) at activation temperature $T = -12$ °C. Source data are provided as Source Data file.

PBAP types II, III and IV correlated with INP concentrations for activation temperatures −10 to −17.5 °C. The highest correlations were found for INP activated at $T = -15$ °C ($R = 0.57$, $R = 0.55$ and $R = 0.64$ for types II–IV, respectively). This INP activation temperature was previously found to be associated with primary biological activity in the Arctic[33,53]. Our analysis strengthens the identification of PBAP by the MBS and links them to high-temperature INP. It also demonstrates the strength of combining online (MBS) and offline (PBAP tracers) techniques.

To summarize, PBAP level and seasonality were correlated with both the high-temperature INP concentrations (specially at $T = -15$ °C) and the proteinaceous fraction of INP. The latter correlation suggests that most of the high-temperature INP were of biological origin and possibly represented by the observed PBAP since concentrations were similar. Hence, we provide a direct observational link between PBAP and high-temperature INP in the Arctic.

## Methods
### Campaign overview
The 'Ny-Ålesund aerosol and cloud experiment' (NASCENT 2019–2020) field experiment took place between 2019 and 2020 in Ny-Ålesund on the Norwegian archipelago of Svalbard. The experiment aimed to study the role of aerosol physio-chemical properties in the formation of clouds in the Arctic. A detailed overview of this large international effort is given by ref. 68. NASCENT partly overlapped in time with the MOSAiC campaign[69].

Intensified observations were conducted in parallel to routine air observations at the Zeppelin Observatory, a joint Global Atmospheric Watch (GAW), EMEP and AMAP Observatory located on the Zeppelin Mountain, about 2 km south–west from the village of Ny-Ålesund and 474 m above sea level. The observatory has been the base for decades of aerosol observations in the Arctic (see review by ref. 70).

During the NASCENT campaign, a MBS and a TEM filter collector were installed behind a whole-air inlet (sampling all particles, including cloud particles up to 40 μm at wind speeds up to 20 ms$^{-1}$[71]), and two INP filter samplers behind PM$_{10}$ inlets. Both inlets were slightly heated to prevent freezing. Both inlets meet the guidelines of the World Meteorological Organization GAW program for aerosol sampling. The aerosols were collected under dry conditions (RH = 23 ± 2%, mean ± standard deviation for the year 2020).

### Sampling and identification of bioaerosols
The multiparameter bioaerosol spectrometer (MBS, University of Hertfordshire, UK)[34] is an instrument designed to assess the possible biological origin of aerosols by measuring UV-induced fluorescence spectra, particle size and (optical) morphology on a single particle basis. Particle differentiation is mainly based on the measured emission spectra using an eight-channel fluorescence spectrometer (covering the wavelength range of 305–655 nm) with an excitation wavelength of 280 nm using a xenon flash lamp. The instrument also retrieves the optical diameter and scattering of the particle, the latter by two linear parallel detector arrays, which gives information on the morphology of the particle. A more in-depth description of the instrument is given by ref. 34. Although MBS sampling already started mid-June 2019, we will focus on the year 2020, where MBS sampled in parallel to collection of INP and TEM samples, except for only a few

short interruptions. The instrument flow was 2.03 L min$^{-1}$, of which 0.33 L min$^{-1}$ was sample flow, and the rest was diverted as bleed and sheath flow.

In the work presented here, we follow our previous classification of fluorescent particles using a decision tree method[39] based on work by ref. 72. In summary, each detection channel is treated individually and assigned two threshold values. The first classification denotes particles where the fluorescence signal is 3 times the background signal and are called fluorescent at the respective channel. The second classification denotes particles where the fluorescence signal is 9 times the background signal; these particles are called highly fluorescent particles at the respective channel. Particles that exhibit highly fluorescent behavior are assigned a spectral class, which represents the channels where their threshold is exceeded. Each spectral class is represented by a combination of letters (A−H) representing their spectral signal (e.g., a $\overline{BCD}$ particle surpassed the second threshold in channels 2, 3, and 4).

Particles that are highly fluorescent and whose highest signal is in channel 2 (or B, central wavelength 364 nm) are classified as fluorescent PBAP and the rest are grouped as highly fluorescent particles (HFP). Particles that are fluorescent but not highly fluorescent at any given channel are grouped as fluorescent particles (FP). This classification is based upon and further described in ref. 39 along with concentration calculations. In summary, the MBS's channel B was designed to narrowly detect the fluorescence of tryptophan, an amino-acid widely found in microorganisms[40]. Within ref. 39, we confirmed this by isolating microorganisms' signal from sea spray aerosol using controlled experiments. For simplicity, we refer to the four most populous PBAP spectral classes ($\overline{B}$, $\overline{BC}$, $\overline{ABC}$, $\overline{ABCD}$) as PBAP types I to IV respectively, and the remaining less prominent spectral classes grouped as PBAP type V. The fluorescence spectra of each type are shown in Fig. S7.

### Ice nucleating particle analysis

Two different sampling strategies were used for the INP analysis, both relying on collection of aerosol filter samples subjected to later analysis in the laboratory.

The first method (Method I) uses weekly filter samples taken behind a PM$_{10}$ inlet installed at the Zeppelin Observatory. The sampling for measuring INP active in the immersion mode were performed continuously and sequentially from Sunday (00:00 UTC) to Saturday (23:59 UTC) using a 10-line Global Sampler (GS-10N, Tokyo Dylec Corp., Japan). Each sample has been collected on a precleaned Whatman Nuclepore track-etched membrane filter (47 mm in diameter and 0.2 µm in pore size) at a flow rate of 3 L min$^{-1}$. We typically extracted aerosol particles collected on the half-cut filters into Milli-Q purified water (≥18 MΩ cm) with a volume of 3.67 ml in a centrifuge tube and then placed the particle-containing droplets with a volume of 5 µm on an aluminum plate coated with a thin layer of Vaseline (petroleum jelly) using an Eppendorf pipette. The droplets were cooled at a cooling rate of 1 °C min$^{-1}$ in the Cryogenic Refrigerator Applied to Freezing Test (CRAFT)[73], and the numbers of the frozen and unfrozen droplets were counted every 0.5 °C. Based on the results of the drop-freezing experiments performed using the CRAFT system, we quantified the number concentration of atmospheric INP active at given temperatures[63,74].

The second method (Method II) is based on weekly filter (Quartz fiber) sampling downstream a PM$_{10}$ inlet for the period of January 2017 to January 2021. For the INP analysis, 72 punches (2 mm diameter) were cut from aliquots (3 cm$^2$) in a laminar flow hood. Each punch was placed in an Eppendorf safelock tube (0.5 ml) with molecular grade water (0.1 mL) (Sigma Aldrich, W4502-1L). The tubes were fixed in a plate and placed into a cold bath, which was cooled from −6 °C to −15 °C in steps of 1 °C for INP detection. Each step consisted of a cooling time of 2 min plus 1 min

at the target temperature before the number of frozen tubes was counted. The plate with the tubes was then placed in a warm water bath (60 °C) for 10 min to deactivate the most temperature sensitive INP and then reanalyzed for INP in the cold bath. Finally, the tubes were placed for 10 min in hot water (95 °C) before being analyzed a last time for INP. The procedure took 2.5 h for each sample and was carried out without interruption. The fraction of INP deactivated by the two heat treatments together is termed the INP proteinaceous fraction. The purpose of the two-step heat treatment was to roughly distinguish between the bacterial and fungal contribution to the total fraction of proteinaceous INPs. Bacterial INP are typically deactivated at 60 °C, whereas some fungal INP withstand 60 °C but are deactivated at 95 °C. An annual cycle (averaged over the 4 years) for the bacterial, fungal and proteinaceous fraction is shown in Fig. S9 along with INP concentration. All four years taken together, it seems like the fungal fraction might be larger (52%, IQ range 39–65%) than the bacterial fraction (mean 29%, IQ range 13–47%). This analysis was based on the work by ref. 75. We chose to work with the proteinaceous fraction only.

Both INP methods were inter-compared and this is shown in Fig. S10 in the SI for $T = -12$ °C. Despite sampling behind two different PM$_{10}$ inlets, using two different methods and distinct sampling schedules, both methods agree well both in absolute numbers and in the annual cycle.

### Transmission electron microscopy

An aerosol impactor sampler (AS-24W, Arios Inc., Tokyo, Japan) was used to collect coarse mode aerosol particles with 50% of a lower cutoff size (>700 nm) in aerodynamic diameter with a flow rate of 1 L per min. These TEM samples were collected on 200 mesh Cu grids with a formvar carbon substrate (U1007, EM-Japan, Japan). The sampling details using the same sampler and location have been reported[76] but for fine-mode samples (<700 nm). The TEM grids were measured using a transmission electron microscope (JEM-1400, JEOL, Japan) with an energy-dispersive X-ray spectrometer (EDS; X-Max 80 mm, Oxford Instruments, Japan). Compositions of nearly all coarse-mode particles collected on two TEM grids from August 7, 2022, were manually measured ($n = 295$ in total). PBAP were classified based on their carbonaceous compositions with a presence of $P$ (>0.4 wt.%), which is a key element of PBAP[77]. Other essential elements of biological species, such as S, Cl, and K, were also commonly detected from PBAP. TEM images were obtained from all PBAP to confirm their characteristic shapes (Figs. S11, S12 and S13).

### Auxiliary parameters

The concentration of eBC was determined using a multi-angle absorption photometer (MAAP, Model 5012, Thermo Fisher Scientific Inc., Germany), which is a filter-based instrument measuring the change in light attenuation as particles are deposited onto the said filter. The light absorption coefficient (abs) is then corrected for artifacts generated by the scattering of particles (also from the filter matrix itself). The site-specific mass absorption cross section of ref. 78 of 10.6 m$^2$g$^{-1}$ was used for the conversion from abs (Mm$^{-1}$) to eBC (ng m$^{-3}$). In addition, we accounted for the wavelength differences of the MAAP reported by ref. 79 by multiplying all eBC values with 1.05. More technical details can be found in ref. 80.

Particulate inorganic and organic concentrations were downloaded from the EBAS database infrastructure (http://ebas.nilu.no/), which were measured using the following techniques: potassium (K$^+$, ion chromatography, IC), sodium (Na$^+$, IC), magnesium (Mg$^{2+}$, IC), calcium (Ca$^{2+}$, IC), iron (Fe, inductively coupled plasma mass spectrometry, ICP-MS), aluminum (Al, ICP-MS), organic carbon (OC, thermal-optical analysis, TOA), levoglucosan (ultra-high performance liquid

chromatography - mass spectrometry, UHPLC-MS), 2-methylerythritol (UHPLC-MS), mannitol (UHPLC-MS) and arabitol (UHPLC-MS).

Radiation measurements (surface albedo, short wave downwelling and upwelling and air temperature) were measured at the BSRN site in Ny-Ålesund[81].

## Trajectories

The normalized difference vegetation index data[82] was derived from Terra Moderate Resolution Imaging Spectroradiometer (MODIS) Vegetation Indices (MOD13Q1) Version 6.1[83]. The data has a time and spatial resolution of 16 days and 250 meters, respectively. An area of 200 (S-N) × 100 km (E−W) was selected around Ny-Ålesund to retrieve average values. Wherever the NVDI value reached a threshold of 1000, it was considered a green pixel. The maximum number of green pixels reached within the area was used to normalize the annual set of observations.

The Hybrid Single-Particle Lagrangian Integrated Trajectory model (HYSPLIT)[84] is used to track the history of air parcels arriving at the observatory. An ensemble of 27 back trajectories were initialized every hour at a height of 250 m. Global Data Assimilation System (GDAS) 1° x 1° archive data was used for the meteorological fields. For surface type exposure, a length of 5 days was chosen. SSMIS-derived 25 km grid resolution sea ice data is taken from EUMETSAT OSI SAF. For each endpoint along the back trajectories, the surface type (i.e., sea-ice, land and open ocean) and whether it resides within the mixed-layer is ascertained. The total time spent over each respective surface type is integrated along each set of ensemble back trajectories. Active fire detection was retrieved from the MODIS Collection 6 Hotspot / Active Fire Detections MCD14ML database distributed by NASA's FIRMS programme[43].

## Data availability

The MBS, MAAP and INP method II data along with TEM images are available are available at Freitas et al.[85]. The data for INP method I data is available at the ADS repository[86]. The organic and inorganic particulate matter concentration data are reported to the EMEP monitoring program and are available from the EBAS database infrastructure (http://ebas.nilu.no/) hosted at NILU. The observatory meteorological data is available at the EBAS database (http://ebas.nilu.no/). The MODIS NDVI data is available on the LP DAAC repository (https://lpdaac.usgs.gov/products/mod13q1v061/). The surface radiation and meteorological data are available in the PANGAEA database[87]. HYSPLIT and GDAS data are available in the ARL archive (https://www.ready.noaa.gov/archives.php). Forest fire data are available on the FIRMS website (https://firms.modaps.eosdis.nasa.gov/download/). Source data are provided with this paper.

## Code availability

The code (using open-source programming language Python, version 3.7.10) used for data processing, data analysis and plotting is available at https://doi.org/10.5281/zenodo.8277139. All packages (e.g., Pandas, NumPy) used are available within the open-source anaconda distribution (available at https://www.anaconda.com/download). Back trajectories were calculated using the HYSPLIT model through the open-source PySplit package (https://pypi.org/project/PySPLIT/).

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

## Acknowledgements

This research was supported by the Swedish Research Council (grant no. 2018-05045, P.Z.), the Knut och Alice Wallenbergs Stiftelse (ACAS project grant no. 2016.0024, R.K.), the Swedish Environmental Agency (Naturvårdsverket) and funding agency FORMAS. This project has received funding from the European Union's Horizon 2020 research and innovation program under Grant Agreement 821205 (FORCeS) (P.Z.). This project has received funding from the European Union's Horizon 2020 research and innovation program under grant agreement no. 101003826 via project CRiceS (Climate Relevant interactions and feedbacks: the key role of sea ice and Snow in the polar and global climate system) (P.Z.), the Environment Research and Technology Development Fund 2–2003 (JPMEERF20202003, K.A. and Y.T.) and 2–2301 (JPMEERF20232001, K.A. and Y.T.) of the Environmental Restoration and Conservation Agency, the JSPS KAKENHI (JP19H01972, Y.T.), and the Arctic Challenge for Sustainability II (ArCS II) Project (JPMXD1420318865, K.A. and Y.T.). Funding to establish organic tracers time series used in the present study were provided by the Norwegian Ministry of Climate and Environment and are gratefully acknowledged. We also acknowledge the Svalbard Integrated Arctic Earth Observing System (SIOS) for their support. The authors thank research engineers Tabea Henning, Ondrej Tesar, Kai Rosman and Birgitta Noone from ACES and the staff from the Norwegian Polar Institute (NPI) for their on-site support. NPI is recognized for its substantial long-term support in maintaining measurements at the Zeppelin Observatory. The authors are grateful for the support and collaboration with the University of Hertfordshire related to the development and support of the MBS instrument (Paul Kaye and Warren Stanley).

## Author contributions

G.P.F., F.C., K.A., K.E.Y., R.K., Y.T. and P.Z. performed measurements and data analysis. G.P.F. performed main data analysis and was the lead author of the manuscript. D.H.-R. performed back trajectory calculations. P.Z. designed the study. All authors contributed to the writing and commenting of the manuscript.

## Funding

## Competing interests

The authors declare no competing interests.
