## [Peer Review File · Nature Communications]

Reviewer #1 (Remarks to the Author):

This is a noteworthy study which for the first time quantitatively links the seasonal cycle of Arctic PBAP emissions to INP and climatic feedback using a novel integrative multi-annual measurement-observation approach based at the Svalbard Arctic mountain site. The impact of these results will be significant to our understanding of the warming Arctic climate and for improving climate feedback models to assess their current uncertainties.

A key component of this study is the integration of 1) real-time multi-parameter UVLIF bioaerosol spectrometer (MBS) observations to successfully differentiate coarse mode fluorescent primary biological aerosol particles (PBAP) from other fluorescent particle classes and hence confirm their sources and response to meteorological factors; and 2) to quantitatively link the different PBAP, proteinaceous terrestrial and oceanic aerosol classes with continuous ice nuclei particle (INP) [CRAFT system] concentration measurements over 4 years; and 3) relevant supporting aerosol parameters including transmission electron microscopy (TEM), standard back trajectory analysis, surface change characteristics and meteorological drivers as well as well cited previous measurement studies at Svalbard.

Particularly convincing are the seasonal "warm temperature" INP measurements linked to relative PBAP class observations with surface change. This quantitative analysis has been missing from previous studies and the authors should be commended for their efforts.

The discrimination and identification of different, co-emitted, proteinaceous particle sources is carefully supported by other relevant measured aerosol components which is one of the significant uncertainties in these measurement approaches. However, they have been careful to acknowledge these uncertainties and "probable" nature of particle types and together with the supporting correlations and arguments, provide convincing evidence for their conclusions and the importance of their work regarding potential impacts for current and future Arctic amplification processes.

Of particular interest would be the provision of a real-world INP vs temperature parameterisation from their terrestrial bioaerosol observations (for different classes) as these are woefully lacking in the literature, particularly for the Arctic region. Whilst there are issues with INP measurement techniques, e.g. the type of filter sample and extraction technique used, which can result in significant activation efficiencies for some temperature ranges the long-term data and the correlations with PBAP etc are very convincing.

To conclude this study does indeed provide the first direct observational link between PBAP and warm INP in the warming Arctic for a range of conditions and the authors should be congratulated for their work.

One minor comment: the Crawford et al. 2017 citation should be corrected.
Crawford, I. et al. Real Time Detection of Airborne Bioparticles in Antarctica. *Atmospheric Chemistry and Physics, Discussions* 1–21 (2017).

It should be:

Crawford, I. et al.: Real-time Detection of Airborne Fluorescent Bioparticles in Antarctica, *Atmos. Chem. Phys.*, 17, 14291–14307, <https://doi.org/10.5194/acp-17-14291-2017>, 2017.

Reviewer #2 (Remarks to the Author):

Review of "Regionally sourced bioaerosols drive high-temperature ice nucleating particles in the Arctic" by Freitas et al., submitted to *Nature Communications*

The manuscript is very well written and covers an important topic, namely, a better characterization of larger size Arctic atmospheric aerosol particles. These particles may be of biogenic origin and, at the same time, may act as ice nucleating particles (INP), as which they influence clouds and radiative processes, and with that weather and climate.

As such, the study will be highly interesting to a more specific audience, namely biologists and atmospheric researchers studying INP who are trying to pinpoint the nature of INP. But I also expect that it will be interesting for a much broader audience, considering the far reaching implications touched on above.

More specifically, in this really good study, larger aerosol particles arriving at the Zeppelin measurement station on Svalbard were examined. One instrument was used for characterizing coarse mode particles (CP), fluorescent particles (FP) and highly fluorescent particles (HFP). The latter group was again divided, forming a group of its own from particles showing a specific characteristic. These were denoted as primary biological aerosol particles (PBAP). Together with data on chemical composition, with backward-trajectories and with data on ice nucleating particles (both, temperature spectra of their concentration and their proteinaceous fraction), the authors derive a number of interesting and important statements and conclusions, which are all formulated in such a way that I can agree with, besides for issues mentioned in my comments below.

Overall, this study will help guide research into further directions, concerning topics such as INP and their sources, biological particles and contributions to large particles from sea spray and biomass burning.

Nevertheless, I did find a number of issues, most often likely relating to the brevity with which the text needed to be formulated. Below I ask for a number of specific additions, including crucial points to which corrections need to be made. In turn, I recommend that chapter 5 should be moved to the supplemental information. Also, I did not check all literature citations, but I checked a few and found two which seemed misplaced and a newer one that should be added (for all, see comments below).

Overall, I can recommend publication of this work, once the comments will have been addressed.

Comments, both major and minor issues:

Remark: Below, references to more than one line typically refer to complete sentences. If only a single line number is given, and it may not be clear what the comment refers to, the respective text is given in the comment.

Lines 37-38: Mentioning "larger size, morphology and composition" in this context is misleading. It has been known for some time, that ice activity from microorganisms originates from macromolecules (Pummer et al., 2015). These can be proteins or polysaccharides. Particularly mentioning the "larger size" is rather misleading, as this larger size rather means shorter atmospheric residence times, decreasing potential atmospheric concentrations.

Pummer, B. G., C. Budke, S. Augustin-Bauditz, D. Niedermeier, L. Felgitsch, C. J. Kampf, R. G. Huber, K. R. Liedl, T. Loerting, T. Moschen, M. Schauperl, M. Tollinger, C. E. Morris, H. Wex, H. Grothe, U. Pöschl, T. Koop, and J. Fröhlich-Nowoisky (2015), Ice nucleation by water-soluble macromolecules, *Atmos. Chem. Phys.*, 15, 4077–4091, doi:10.5194/acp-15-4077-2015.

Line 62: You are mentioning times with elevated concentrations of the biomass levoglucosan here, pointing to Figs. S1-b and S3-a. But these times with elevated concentrations cannot be seen in Fig. S1-b. You could mark the background in this figure for times when levoglucosan exceeds a certain value.

In the following lines, you refer to a connected influence from active forest fires. Figure S4-S6 indicate, which days you may refer to when you refer to these biomass burning aerosols, but that also is not totally clear from these figures: back-trajectories cover the fire data, so specifically for July 7, an influence is difficult to see. Also, are there explanations for elevated levoglucosan concentrations in late January and in February?

Line 67: You are referring to Fig. S3-a. But I don't know how to see the relation of HPF to long-range transport or levoglucosan in this figure, or elsewhere, if I am not mistaken.

Lines 70-71: Did you do a correlation between eBC and HFP for the first months alone? There seems to be a trend with HPF following eBC also in the first 4 months, with both parameters showing a downward trend from January to end of April by roughly one order of magnitude.

There then is a switch and the correlation is different in the second half of the year - which might point towards different sources for the particles for the first and second half of the years - maybe discriminating times with influence from Arctic haze from those without. (Provided the instruments always worked similarly well.) This switch causes low R values when correlating all months simultaneously.

Therefore I suggest to examine a possible correlation for the first 4 months separately, and if it is there, include it in your study.

Line 73: Indeed, it is striking that both HPF and PBAP show their highest concentrations for the lowest wind speeds. This is worth mentioning here.

Lines 74-76: The sentence came somewhat surprising to me. Do you expect that there was a continuous biomass burning influence throughout the second half of the year? If you do not assume this, reframe this sentence, pointing explicitly to the episodes when HPF were high during biomass burning. For that it would help to mark the periods with high levoglucosan, e.g. in the background of Fig. 1-a.

Line 76-77: "MBS is capable of differentiating biomass burning particles from PBAP." The reasoning behind this completely eluded me, and it only became somewhat clearer after I had read the methods section. For your assumption, one at least needs to know that you divided all highly fluorescent particles into PBAP and HFP. Mention this here in the text at least briefly. And a few more words on the basic ideas behind your statement here would be good.

Line 87: Same as for HPF - clearly, concentration were highest for lowest wind, and that may mean something (particularly concerning local terrestrial emissions) and should be mentioned / discussed.

Line 90: "did not correlate with long-range transport air masses of continental origin." Refer the reader to Fig. S3-b, or delete this statement here, as you discuss it more in the next paragraph. And also, where can this be seen? Fig. S3-b shows higher concentrations of PBAP for the "more terrestrial" datapoints.

Line 94-95: Concerning Fig. S3: There is no Figure S3-c, so I wonder if this figure is missing, or if you meant Fig. S3-b instead. However, Fig. S3-b seems to imply a rather terrestrial origin for all, CP, FP, HFP and PBAP (they all have their highest concentrations for the most terrestrial conditions). This figure is not supporting some of the statements you make, and / or needs to be explained in more detail.

Caption of Fig. 2: Maybe point out that you only report air temperatures above 0°C.

Lines 114-116: There is a study that was published a short while ago, Sze et al. (2023), on INP data obtained in Northern Greenland during the course of two years. Their results on INP are comparable to yours, in that they show the same annual INP cycle with high concentrations in summer for the whole examined time period. They also report very similar biogenic fractions obtained upon heating, throughout the year. This study should be included here and maybe at other suitable locations.

Sze, K. C. H., H. Wex, M. Hartmann, H. Skov, A. Massling, D. Villanueva, and F. Stratmann (2023), Ice-nucleating particles in northern Greenland: annual cycles, biological contribution and parameterizations, *Atmos. Chem. Phys.*, 23, 4741–4761, doi:10.5194/acp-23-4741-2023.

Lines 119-120: As far as I saw, the information you cite here is not given in publication, Jensen et al. (2022), which you are citing here. They examined samples from spring 2015 and summer 2016. Therefore, this sentence is highly misleading. Instead, the abovementioned new publication by Sze et al. (2023) does describe a similar pattern for a period of two years.

Line 140: Concerning the positively correlated concentrations of CP and INP, I suggest to discuss at least shortly that different inlet cut-offs were used for MBS and filter sample collection, and how that could influence the results.

Line 151-152: "INP temperatures" - More precisely this should be "INP concentrations at different temperatures".

Line 152: FP also correlated with levoglucosan, albeit less strongly than HFP. This may be worth mentioning.

Line 171, Chapter 5 on Microscopy analysis: Certainly, a lot of work was invested in this, but it does not help the main conclusions of the study. Therefore, I suggest to move this whole chapter, including all text and the figure, to the supplemental information, which then leaves room to add the few short additional explanations suggested in this review to the main part of the text.

Line 174 and again line 279: "case study of the two samples": Why only these two, and why then mention six samples in Table S1? Homogenize!

Line 181: "lower right corner of panel G in Figure S9": It is unclear what you are referring to here. Do you mean the "shadow" in the upper right corner of Fig. S9, panel G?

Lines 186-188: I tried to find the here given information in the cited publication, but only found: "Recent studies corroborate our findings whereby INPs are $>3 \mu\text{m}$ in Arctic aerosol".

On the other hand, Creamean et al. (2018) had higher INP concentrations in the size range $>3 \mu\text{m}$, compared to the one from 1.2-3 μm .

It is therefore needed that you check and revise this statement.

Creamean, J. M., R. M. Kirpes, K. A. Pratt, N. J. Spada, M. Maahn, G. de Boer, R. C. Schnell, and S. China (2018), Marine and terrestrial influences on ice nucleating particles during continuous springtime measurements in an Arctic oilfield location, *Atmos. Chem. Phys.*, 18, 18023–18042, doi:10.5194/acp-18-18023-2018.

Line 235: "three instruments": You are certainly referring to MBS, TEM and INP? This is confusing as filter collection is not an instrument and you likely rather refer to the methods. Please specify / revise.

Lines 245-246: Although this is certainly described in Freitag et al., at least shortly mention what is the background for this characterization of PBAP, as it is a crucial point for your analysis.

Line 252: "Two different sampling strategies": It is unclear, which of the above reported data were obtained from which of these two methods.

When was the first method used, when was the second methods used? Were both methods compared, and if yes, what resulted from this?

Line 259: "particle-containing droplets": Specify the volume of water used to wash off the filters and of the particle containing droplets.

Lines 271-272: You mention that the two heat treatments together were used to determine the proteinaceous fraction. How? Was the heat treatment at 60°C used for anything on its own? If not, why was it done?

Caption Figure S3: Panel A has a gray bar in late March – what does that bar imply?

Caption Figure S4: “where only mixing-layer (ML) was considered”: How do you define the mixing-layer?

Reply to reviewers: Freitas et al., “Regionally sourced bioaerosols drive high-temperature ice nucleating particles in the Arctic”

We would like to thank both reviewers for their comments and suggestions for improvements, which have helped to increase the quality of our manuscript. Our responses to the specific comments follow in blue, and changes translated to the main text are shown in red.

1 Reviewer 1

This is a noteworthy study which for the first time quantitatively links the seasonal cycle of Arctic PBAP emissions to INP and climatic feedback using a novel integrative multi-annual measurement-observation approach based at the Svalbard Arctic mountain site. The impact of these results will be significant to our understanding of the warming Arctic climate and for improving climate feedback models to assess their current uncertainties.

A key component of this study is the integration of 1) real-time multi-parameter UVLIF bioaerosol spectrometer (MBS) observations to successfully differentiate coarse mode fluorescent primary biological aerosol particles (PBAP) from other fluorescent particle classes and hence confirm their sources and response to meteorological factors; and 2) to quantitatively link the different PBAP, proteaceous terrestrial and oceanic aerosol classes with continuous ice nuclei particle (INP) [CRAFT system] concentration measurements over 4 years; and 3) relevant supporting aerosol parameters including transmission electron microscopy (TEM), standard back trajectory analysis, surface change characteristics and meteorological drivers as well as well cited previous measurement studies at Svalbard. Particularly convincing are the seasonal "warm temperature" INP measurements linked to relative PBAP class observations with surface change. This quantitative analysis has been missing from previous studies and the authors should be commended for their efforts.

The discrimination and identification of different, co-emitted, proteinaceous particle sources is carefully supported by other relevant measured aerosol components which is one of the significant uncertainties in these measurement approaches. However, they have been careful to acknowledge these uncertainties and "probable" nature of particle types and together with the supporting correlations and arguments, provide convincing evidence for their conclusions and the importance of their work regarding potential impacts for current and future Arctic amplification processes.

Of particular interest would be the provision of a real-world INP vs temperature parameterisation from their terrestrial bioaerosol observations (for different classes) as these are woefully lacking in the literature, particularly for the Arctic region. Whilst there are issues with INP measurement techniques, e.g. the type of filter sample and extraction technique used, which can result in significant activation efficiencies for some temperature ranges the long-term data and the correlations with PBAP etc are very convincing.

To conclude this study does indeed provide the first direct observational link between PBAP and warm INP in the warming Arctic for a range of conditions and the authors should be congratulated for their work.

We thank the reviewer for their positive review. We agree and hope that our work will help develop more real-world INP vs temperature parameterizations in future work. Providing this kind of parameterization would be beyond the scope of our work, however, our large amount of data will be publicly available for other scientists or groups who are specialized in developing INP parameterizations.

- One minor comment: the Crawford et al. 2017 citation should be corrected. Crawford, I. et al. Real Time Detection of Airborne Bioparticles in Antarctica. Atmospheric Chemistry and Physics, Discussions 1–21 (2017). It should be: Crawford, I. et al.: Real-time Detection of Airborne Fluorescent Bioparticles in Antarctica, Atmos. Chem. Phys., 17, 14291–14307, <https://doi.org/10.5194/acp-17-14291-2017>, 2017.

The reference has been updated.

2 Reviewer 2

Review of “Regionally sourced bioaerosols drive high-temperature ice nucleating particles in the Arctic” by Freitas et al., submitted to Nature Communications

The manuscript is very well written and covers an important topic, namely, a better characterization of larger size Arctic atmospheric aerosol particles. These particles may be of biogenic origin and, at the same time, may act as ice nucleating particles (INP), as which they influence clouds and radiative processes, and with that weather and climate.

As such, the study will be highly interesting to a more specific audience, namely biologists and atmospheric researchers studying INP who are trying to pinpoint the nature of INP. But I also expect that it will be interesting for a much broader audience, considering the far reaching implications touched on above.

More specifically, in this really good study, larger aerosol particles arriving at the Zeppelin measurement station on Svalbard were examined. One instrument was used for characterizing coarse mode particles (CP), fluorescent particles (FP) and highly fluorescent particles (HFP). The latter group was again divided, forming a group of its own from particles showing a specific characteristic. These were denoted as primary biological aerosol particles (PBAP). Together with data on chemical composition, with backward-trajectories and with data on ice nucleating particles (both, temperature spectra of their concentration and their proteinaceous fraction), the authors derive a number of interesting and important statements and conclusions, which are all formulated in such a way that I can agree with, besides for issues mentioned in my comments below.

Overall, this study will help guide research into further directions, concerning topics such as INP and their sources, biological particles and contributions to large particles from sea spray and biomass burning.

Nevertheless, I did find a number of issues, most often likely relating to the brevity with which the text needed to be formulated. Below I ask for a number of specific additions, including crucial points to which corrections need to be made. In turn, I recommend that chapter 5 should be moved to the supplemental information. Also, I did not check all literature citations, but I checked a few and found two which seemed misplaced and a newer one that should be added (for all, see comments below).

Overall, I can recommend publication of this work, once the comments will have been addressed.

We thank the reviewer for their positive and constructive review. We have addressed the comments (see reply below) which increased the clarity of our manuscript. As suggested, we have moved chapter 5 to the SI. However, we added a few of the TEM images to Figure 1, because we believe that they should be shown to the reader in the main text (the TEM analysis is an additional key technique that gave further confidence on the detected PBAPs which e.g. is already mentioned in the abstract).

Comments, both major and minor issues:

Remark: Below, references to more than one line typically refer to complete sentences. If only a single line number is given, and it may not be clear what the comment refers to, the respective text is given in the comment.

- Lines 37-38: Mentioning "larger size, morphology and composition" in this context is misleading. It has been known for some time, that ice activity from microorganisms originates from macromolecules (Pummer et al., 2015). These can be proteins or polysaccharides. Particularly mentioning the "larger size" is rather misleading, as this larger size rather means shorter atmospheric residence times, decreasing potential atmospheric concentrations.

Pummer, B. G., C. Budke, S. Augustin-Bauditz, D. Niedermeier, L. Felgitsch, C. J. Kampf, R. G. Huber, K. R. Liedl, T. Loerting, T. Moschen, M. Schauperl, M. Tollinger, C. E. Morris, H. Wex, H. Grothe, U. Pöschl, T. Koop, and J. Fröhlich-Nowoisky (2015), Ice nucleation by water-soluble macromolecules, *Atmos. Chem. Phys.*, 15, 4077–4091, doi:10.5194/acp-15-4077-2015.

We agree and have modified the sentence Whilst also including Pummer *et al.* 2015 as a reference to the statement (no. 30):

"This is due to their larger size, morphology and composition and especially their excretion of ice nucleating proteins macromolecules^{29,30}."

- Line 62: You are mentioning times with elevated concentrations of the biomass levoglucosan here, pointing to Figs. S1-b and S3-a. But these times with elevated concen-

trations cannot be seen in Fig. S1-b. You could mark the background in this figure for times when levoglucosan exceeds a certain value.

Figure S1 (Here as A1) in the supporting information has been updated with a new panel (panel S1-c) as suggested to support these statements. The panel clearly shows that during summer and fall, an increase in levoglucosan concentrations increases the contribution of fluorescent particles to coarse particles.

Figure A1: **Timelines of parameters measured during the year 2020 of the NASCENT campaign.** a) Coarse mode concentration as measured by the MBS and sodium concentration in filter samples. b) Fluorescent ($\sigma > 3$) particle concentration and wind speed. c) Contribution of fluorescent particles to coarse particles and levoglucosan particulate concentration.

In the following lines, you refer to a connected influence from active forest fires. Figure S4-S6 indicate, which days you may refer to when you refer to these biomass burning aerosols, but that also is not totally clear from these figures: back-trajectories cover the fire data, so specifically for July 7, an influence is difficult to see. Also, are there explanations for elevated levoglucosan concentrations in late January and in February?

Figures S4-S6 were modified so that air trajectories and active fires are shown in two separate panels. Panel (b) in each figures shows active forest fires. The number of active forest fires encountered by the trajectories is now explicitly mentioned in the figure captions. In the previous version of our work, the elevated levoglucosan concentrations in January and February were not thoroughly investigated as they took place outside of our season of interest (summer). However, the surface contribution to each back-trajectory (see Figure S3-a) link these periods to probably terrestrial sources, indicating possibly anthropogenic sources. To clarify, we added to the revised manuscript the following sentence:

"The levoglucosan concentrations were as well elevated in January and February (see Figure S3-a) most likely due to transport of biomass burning aerosol from the continents⁴⁵."

- Line 67: You are referring to Fig. S3-a. But I don't know how to see the relation of HPF to long-range transport or levoglucosan in this figure, or elsewhere, if I am not mistaken.

For further clarification we have updated panel (a) from Figure 1 (in the main text, A2 here) to now contain a third spine with the levoglucosan particulate mass which shows the resemblance to the HPF concentration.

- Lines 70-71: Did you do a correlation between eBC and HFP for the first months alone? There seems to be a trend with HPF following eBC also in the first 4 months, with both parameters showing a downward trend from January to end of April by roughly one order of magnitude.

There then is a switch and the correlation is different in the second half of the year - which might point towards different sources for the particles for the first and second half of the years - maybe discriminating times with influence from Arctic haze from those without. (Provided the instruments always worked similarly well.) This switch causes low R values when correlating all months simultaneously.

Therefore I suggest to examine a possible correlation for the first 4 months separately, and if it is there, include it in your study.

After reevaluating the correlation between HFP and eBC which was low throughout the year, we found some flagged values that were mistakenly not removed. These caused incorrect correlation numbers when calculating for the whole year. The correlation is now 0.69 . All correlations numbers were corrected in Figure 3 and the text was modified to describe the actual link between HFP (and other particle groups) and eBC.

The text changed follows:

"At the beginning of the year, the concentrations of equivalent black carbon (eBC, Figure 1-a) were elevated due to the Arctic Haze phenomenon, which describes the accumulation of anthropogenic aerosol from long-range transport in ~~late winter and early spring~~ the first half of the year⁴⁴. ~~The Arctic haze is mainly composed of aged soot particles that generally do not exhibit strong fluorescence.~~ The HFP concentration for this period ~~did not~~ followed the eBC concentration ($R = 0.84$ for January-June) and stayed at around 10^{-1} L^{-1} . For the second half of the year (July-December), ~~there was a clear correlation with eBC ($R = 0.80$ for June-December compared to $R = 0.43$ for the whole year) and~~ the HFP concentration increased to values as high as 10^1 L^{-1} and continued to show a high correlation to eBC ($R = 0.69$)."

- Line 73: Indeed, it is striking that both HPF and PBAP show their highest concentrations for the lowest wind speeds. This is worth mentioning here.

Figure A2: Timelines of several physical, aerosol and INP parameters measured during the year 2020 of the NASCENT campaign. a) Highly fluorescent ($\gg 9\sigma$) particle concentration, equivalent black carbon (eBC) and levoglucosan concentration. b) Primary biological aerosol particle (PBAP) concentration, vegetation index (NVDI), surface albedo and temperature at Ny-Ålesund. Red star indicates the collection of transmission electron microscopy (TEM) grids. c) Spectral classes contribution to PBAP signal. d) Ice nucleating particles concentration at different nucleating temperatures. Panels (e-h) show TEM images of PBAP sampled on the 7th of August, 2020. The remaining PBAP images for this day are shown in Figure S12.

PBAP high concentration for lowest wind speeds is mentioned in the following paragraph. The sentence has been expanded to say:

The concentration of HFP did not show any clear dependence on wind speed in contrast to the concentrations of CP and FP (Figure S2-c). In fact, HFP concentration were higher for lower wind speeds, suggesting that HFP would not have sea spray or mineral dust as primary sources. This might indicate that HFP are either locally produced by non-wind driven mechanisms or long-range transported.

- Lines 74-76: The sentence came somewhat surprising to me. Do you expect that there was a continuous biomass burning influence throughout the second half of the year? If you do not assume this, reframe this sentence, pointing explicitly to the episodes when HFP were high during biomass burning. For that it would help to mark the periods with high levoglucosan, e.g. in the background of Fig. 1-a.

We agree. The text has been modified to express that the influence of biomass burning pertains to only certain events with high levoglucosan concentrations. The new panel (a) of Figure 1 with the added levoglucosan concentration now better illustrates the observed relationship. We added the following sentence to the revised manuscript:

"During the second half of the year, the elevated concentrations of HFP coincided with elevated levoglucosan concentration (Figure 1-a). The long-range transport of biomass burning aerosol⁴⁶ could have carried primary particles containing polyaromatic carbon chains that possess strong fluorescence properties⁴⁷."

- Line 76-77: "MBS is capable of differentiating biomass burning particles from PBAP"

The reasoning behind this completely eluded me, and it only became somewhat clearer after I had read the methods section. For your assumption, one at least needs to know that you divided all highly fluorescent particles into PBAP and HFP. Mention this here in the text at least briefly. And a few more words on the basic ideas behind your statement here would be good.

We agree and the text was modified to briefly explain the difference between HFP and PBAP. The statement was further elaborated. The following text was added/modified:

"FP with high fluorescence (see Methods) were subdivided into highly fluorescent particles (HFP) and fluorescent PBAP. Their differentiation was based on the center wavelength of fluorescence. Those with the main fluorescence signal at 364 nm were classified as PBAP, and all others as HFP. This was based on previous work by Freitas *et al*³⁹. The concentration of HFP highly fluorescent particles (HFP (mean: 0.2 L⁻¹, IQ range: 0.02–0.1 L⁻¹, Figure 1-a) showed distinct peaks with values up to 1 L⁻¹ occurring mainly in summer."

"These findings could indicate that the MBS is capable of differentiating Arctic haze and biomass burning particles from PBAP. This is promising, as interference from non-biological fluorescent particles is a problem often associated with the identification of PBAP using fluorescence-based measurements^{40,48}."

- Line 87: Same as for HPF - clearly, concentration were highest for lowest wind, and that may mean something (particularly concerning local terrestrial emissions) and should be mentioned / discussed.

The discussion concerning the repercussions of the HFP has been developed further:

"The concentration of HFP did not show any clear dependence on wind speed in contrast to the concentrations of CP and FP (Figure S2-c). In fact, HFP concentration were higher for lower wind speeds, suggesting that HFP would not have sea spray or mineral dust as primary sources. This might indicate that HFP are either locally produced by non-wind driven mechanisms or long-range transported."

- Line 90: “did not correlate with long-range transport air masses of continental origin.” Refer the reader to Fig. S3-b, or delete this statement here, as you discuss it more in the next paragraph. And also, where can this be seen? Fig. S3-b shows higher concentrations of PBAP for the “more terrestrial” datapoints.

This sentence was removed and the discussion moved to the next paragraph. As can be seen from Figure S3-b, the difference in PBAP concentration between more terrestrial and more oceanic trajectories is not as pronounced as for the HFP. The text has been modified on the next paragraph to:

"To assess sources, the concentrations of the main particle groups (CP, FP, HFP and PBAP) were analyzed with respect to air mass origin on a normalized scale, where negative fractions signified a terrestrial source and positive fractions an oceanic source. Figure S3-**eb** shows that PBAP were only slightly connected to the back trajectories of more terrestrial regions and CP and FP to more oceanic regions. **However, since the difference between terrestrial and oceanic trajectories is not sufficiently large, reasonable conclusions cannot be drawn for these 3 groups of particles. HFP on the other hand, were clearly connected to terrestrial trajectories, possessing concentrations at least one order of magnitude higher than oceanic trajectories.**"

- Line 94-95: Concerning Fig. S3: There is no Figure S3-c, so I wonder if this figure is missing, or if you meant Fig. S3-b instead. However, Fig. S3-b seems to imply a rather terrestrial origin for all, CP, FP, HFP and PBAP (they all have their highest concentrations for the most terrestrial conditions). This figure is not supporting some of the statements you make, and / or needs to be explained in more detail.

We apologize for this typo. Fig S3-b was supposed to be referenced instead. The text has been modified to emphasize the fact that only HFP had a sufficiently big difference between more terrestrial or more oceanic trajectories. Thus, only a conclusion of HFP can be reasonably drawn. The modified text is shown in the previous comment answer.

- Caption of Fig. 2: Maybe point out that you only report air temperatures above 0°C. This has been added to the figure caption.
"Along with the observed air temperature (**above 0 °C**) at Ny-Ålesund and INP concentration (at T = -12 °C)."
- Lines 114-116: There is a study that was published a short while ago, Sze et al. (2023), on INP data obtained in Northern Greenland during the course of two years. Their results on INP are comparable to yours, in that they show the same annual INP cycle with high concentrations in summer for the whole examined time period. They also report very similar biogenic fractions obtained upon heating, throughout the year. This study should be included here and maybe at other suitable locations.

Sze, K. C. H., H. Wex, M. Hartmann, H. Skov, A. Massling, D. Villanueva, and F. Stratmann (2023), Ice-nucleating particles in northern Greenland: annual cycles,

biological contribution and parameterizations, Atmos. Chem. Phys., 23, 4741–4761, doi:10.5194/acp-23-4741-2023.

See next answer. This study has replaced Jensen et al. 2022 as it is the correct citation.

- Lines 119-120: As far as I saw, the information you cite here is not given in publication, Jensen et al. (2022), which you are citing here. They examined samples from spring 2015 and summer 2016. Therefore, this sentence is highly misleading. Instead, the abovementioned new publication by Sze et al. (2023) does describe a similar pattern for a period of two years.

This was a reference manager mistake as both Jensen et al. 2022 and Sze et al. 2023 work had Greenland in the title. The study supposed to be cited here was indeed Sze et al. 2023. We have corrected this mistake in the revised manuscript.

- Line 140: Concerning the positively correlated concentrations of CP and INP, I suggest to discuss at least shortly that different inlet cut-offs were used for MBS and filter sample collection, and how that could influence the results.

Two sentences were added to address this comment by the reviewer:

"The concentrations of CP and INP were not positively correlated at any nucleation temperature (Figure 3-a, utilizing INP Method I). The MBS sampled behind a whole air inlet, while INP filters were collected behind a PM₁₀ inlet (see Methods). Nonetheless, 99.997% of the CP were sized between 0.8 and 10 μm and thus sampled by both inlets."

- Line 151-152: "INP temperatures" - More precisely this should be "INP concentrations at different temperatures".

The sentence has been adjusted accordingly. It now reads as:

"FP correlated strongly with eBC, sodium and magnesium as seen for CP and had a low correlation with levoglucosan, but not with any of the other organic tracers or INP concentrations at different temperatures, indicating that FP are indeed a subset of sea spray particles and whose contribution to CP is exacerbated during long-range transport events of biomass burning."

- Line 152: FP also correlated with levoglucosan, albeit less strongly than HFP. This may be worth mentioning.

We agree. The sentence has been modified (see reply from previous comment).

- Line 171, Chapter 5 on Microscopy analysis: Certainly, a lot of work was invested in this, but it does not help the main conclusions of the study. Therefore, I suggest to move this whole chapter, including all text and the figure, to the supplemental information, which then leaves room to add the few short additional explanations suggested in this review to the main part of the text.

We have taken the suggestion from the reviewer and moved the whole chapter to the SI. We have modified the text in the main manuscript to account for this change:

"~~However, t~~ The PBAP concentration numbers (10^{-2} – 10^{-1} L $^{-1}$) were quite close to those found by Johansen *et al.*²⁵ using spore traps in Ny-Ålesund. ~~In addition, the MBS-~~retrieved PBAP classification and concentration were corroborated by transmission electronic microscopy (TEM) imaging. Arctic PBAP sampled on the 7th August 2020 (red star in Figure 1-b) are shown in panels e-h of Figure 1. The link between MBS and TEM derived PBAP is described in Section S1 of the SI. The increase in PBAP concentration in June coincided with an increase in ambient air temperature."

The Figure and the text in the SI have also been changed and are shown in the next comment.

- Line 174 and again line 279: “case study of the two samples”: Why only these two, and why then mention six samples in Table S1? Homogenize!

The explanation has been added to the text:

"Transmission electron microscopy (TEM) analysis was applied to coarse particle samples (aerodynamical size $> 1 \mu\text{m}$) that were collected simultaneously on the same sampling line as the MBS. These samples were collected in December 2019 (1 sample), August 2020 (4 samples), and September 2020 (1 sample, Table S1). Here, we present a case study of the two samples collected on August 7, 2020, with a total of 75-minute sampling period. ~~We chose these two samples to have better statistics for comparison. Since they were taken on the same day, they are likely to have been emitted from the same sources. TEM images of PBAP found in other samples are shown in Figure S13.~~ In total, 13 PBAP were classified according to the TEM composition analysis⁷⁸(Figures 1e-h; Table S1 and Figure S12). PBAP identified by TEM analysis represented approximately 4% of all analyzed particles or $\sim 0.2 \text{ L}^{-1}$, which is comparable to the MBS derived PBAP concentration of 0.11 L^{-1} for the same day. An example of the elemental analysis of the particle in panel (a)(e) of Figure 1 in the main manuscript is shown in Figure S11 (panels a-d) ~~panels (d-g)~~. Carbon (da) is concentrated mainly in the hemispheres of the PBAP, while potassium (eb) and phosphorus (fc) are concentrated in small inclusions. Nitrogen is seen throughout the particle (gd). Some particles had structures that resembled pili and flagellum (such as in the lower right corner of panel G in Figure S12). Most PBAP particles had the same slightly ellipsoidal shape, while one particle had a long tail and another was found to be attached to an organic particle (Figures 1-f and g in the main manuscript, respectively). Their equivalent area diameters ranged from 1.5 to $4 \mu\text{m}$, with a diameter mode between 2-3 μm , which compared well with the optical diameter range (1-7 μm) and mode (2-4 μm) of PBAP measured by the MBS (see panel ~~I~~e in Figure S11), demonstrating that both methods independently derived similar sizes. PBAP were probably mostly represented by bacteria or fungal spores according to their shape,

composition, structure, size and presence of flagellum². Arctic INP (T = -20 °C) have been observed in summer to be dominated by particles in the size range 1.2-3 μm, deviating from the rest of the year when larger particles are dominating (3-12 μm, see supplementary Figure 15 of Creamean *et al.* 2022³). Our findings seem to reflect these results.

During the same day, the MBS measured 48 PBAP particles, representing 0.07% of the CP. Their contribution by type per size is shown in Figure S11-hf, revealing the presence of the five PBAP types identified by the MBS during the course of this day. Furthermore, the mean scattering signal (and interquartile) in both linear detector arrays of the MBS is shown in Figure S11-jg as an average of all detected PBAP. This scattering signal resembles that of symmetric particles (mean asymmetry parameter of 24, i.e. the degree of dissimilarity between both linear detector signals on a scale from 1 to 100) and slightly elongated particles (peak-to-mean ratio of 3.05, i.e. the ratio between the most intense pixel and the mean value across each individual array)⁴. Thus, the scattering signal measured by the MBS for the PBAP represented the morphology (symmetrical and slightly ellipsoidal) observed in the TEM images."

Figure A3: TEM images and MBS signal from the 7th of August of 2020. (a-e) TEM images of PBAP particles collected by two samples during the day. (d-g) (a-d) Composition analysis of the PBAP shown at panel (ae) of Figure 1 in the main manuscript. (e) Normalized size distribution of PBAP measured by TEM and of PBAP measured by the MBS. (f) Spectral signature contribution at each size, measured by the MBS. (g) Mean and interquartile scattering signal of PBAP particles by the MBS during the day.

- Line 181: “lower right corner of panel G in Figure S9”: It is unclear what you are referring to here. Do you mean the "shadow" in the upper right corner of Fig. S9, panel G?

We have changed to text to refer to an arrow instead of a place in the panel. We have added arrows in the figures to point to the pili/flagelum.

Figure A4: **Transmission electronic microscopy (TEM) images of bioaerosols. (a-ji)** Different primary biological particles measured by TEM within two samples collected on the 7th of August of 2020. **Red arrow in panel (g) points to a possible flagellum. This figure has been updated by removing one panel (now shown in Figure 1 in the main manuscript) and by adding a red arrow in panel (g) to more clearly point towards the possible flagellum of one of the PBAP.**

Figure A5: **Transmission electronic microscopy (TEM) images of bioaerosols.** Different bioaerosols measured by TEM collected on the 9th of August (a-d) and 26th of August (e,f). **Red arrows point to possible flagellum in the images.**

- Lines 186-188: I tried to find the here given information in the cited publication, but only found: "Recent studies corroborate our findings whereby INPs are $>3 \mu\text{m}$ in Arctic aerosol".

On the other hand, Creamean et al. (2018) had higher INP concentrations in the size range $>3 \mu\text{m}$, compared to the one from 1.2-3 μm .

It is therefore needed that you check and revise this statement.

Creamean, J. M., R. M. Kirpes, K. A. Pratt, N. J. Spada, M. Maahn, G. de Boer, R. C. Schnell, and S. China (2018), Marine and terrestrial influences on ice nucleating particles during continuous springtime measurements in an Arctic oilfield location, *Atmos. Chem. Phys.*, 18, 18023–18042, doi:10.5194/acp-18-18023-2018.

The statement in the text "Arctic INP ($T = -20^\circ\text{C}$) have been observed in summer to be dominated by particles in the size range 1.2-3 μm , deviating from the rest of the year when larger particles are dominating (3-12 μm)" is refereed in text to Creamean *et al.* 2022. Their findings are shown in Figure 15 in their supplementary information (it is a bit hidden). They saw that particles in the size range 1.2-3 μm dominate INP at -20°C for the months of June, July and September of the year 2020. August saw an equal contribution of 1.2-3 μm and 3-12 μm particles. Such findings are not discussed in the main text. Indeed, for the remainder of their measurements 3-12 μm particles dominated the contribution. For clarification, we added the statement ", see supplementary figure 15 of Creamean *et al.* 2022" to the original sentence.

Creamean, J.M., Barry, K., Hill, T.C.J. et al. Annual cycle observations of aerosols capable of ice formation in central Arctic clouds. *Nat Commun* 13, 3537 (2022). <https://doi.org/10.1038/s41467-022-31182-x>

- Line 235: "three instruments": You are certainly referring to MBS, TEM and INP? This is confusing as filter collection is not an instrument and you likely rather refer to the methods. Please specify / revise.

The phrase has been clarified and now reads as:

"Although MBS sampling already started mid-June 2019, we will focus on the year 2020, where ~~all three instruments sampled in parallel~~ MBS sampled in parallel to collection of INP and TEM samples, except for only a few short interruptions. The instrument flow was 2.03 lpm, of which 0.33 lpm was sample flow, and the rest was diverted as bleed and sheath flow."

- Lines 245-246: Although this is certainly described in Freitag et al., at least shortly mention what is the background for this characterization of PBAP, as it is a crucial point for your analysis.

We have further developed the discussion:

"articles that are *highly fluorescent* and whose highest signal is in channel 2 (or B, central wavelength 364 nm) are classified as fluorescent primary biological aerosol particles (PBAP) and the rest are grouped as highly fluorescent particles (HFP). Particles that are *fluorescent* but not *highly fluorescent* at any given channel are grouped as fluorescent particles (FP). This classification is based upon and further described and discussed in Freitas *et al.*³⁹ along with discussion on concentration calculations. In summary, the MBS's channel B was designed to narrowly detect the fluorescence of tryptophan, an amino-acid widely found in microorganisms⁴⁰. Within Freitas *et al.*³⁹, we confirmed this by isolating microorganisms' signal from sea spray aerosol using controlled experiments. For simplicity, we refer to the 4 most populous PBAP spectral classes (\overline{B} , \overline{BC} , \overline{ABC} , \overline{ABCD}) as PBAP types I to IV respectively, and the remaining less prominent spectral classes grouped as PBAP type V. The fluorescence spectra of each type are shown in Figure S7."

- Line 252: "Two different sampling strategies": It is unclear, which of the above reported data were obtained from which of these two methods.

When was the first method used, when was the second methods used? Were both methods compared, and if yes, what resulted from this?

We have clarified the usage of the two different INP methods in the text. The comparison was carried out and Figure S10 (here as A6) was added to the SI along with the following texts:

"The correlation of the identified PBAP with air temperature and vegetation index suggested that most of the PBAP were of local regional terrestrial origin. Two methods were used to retrieve INP concentrations at the site. Furthermore Here, we present strong evidence that PBAP were the main contributor to the concentration of INP active at warmer higher air temperatures."

"The concentration of cold low-temperature INP (utilizing INP Method I, activated at -25 to -30°C, not shown) remained fairly stable throughout the year (from 10^{-1} L^{-1} to 10^1 L^{-1}) and showed no clear seasonal cycle as previously reported by Schrod *et al.*⁵²."

"Over 4 years (2017-2020), we measured the concentration of INP (utilizing INP Method II, at activation temperature -12 °C), as well as their proteinaceous (heat-labile) fraction (Figure 2-a, along with the air temperature)."

"Both INP methods were inter-compared and this is shown in Figure S10 in the SI for $T = -12 \text{ C}^\circ$. Despite sampling behind two different PM_{10} inlets, using two different methods and distinct sampling schedules, both methods agree well both in absolute numbers and in the annual cycle."

- Line 259: "particle-containing droplets": Specify the volume of water used to wash off the filters and of the particle containing droplets.

Figure A6: **Comparison between INP methods I and II for INP active at T = -12 °C.** (a) Using logarithmic scale and (b) linear scale.

We slightly adjusted the experimental setup depending on the sampling condition (e.g., sample volume). For specification, we have modified the sentence to :

"We typically extracted aerosol particles collected on the half-cut filters into Milli-Q purified water ($\geq 18 \text{ M}\Omega \text{ cm}$) with a volume of 3.67 ml in a centrifuge tube and then placed the particle-containing droplets with a volume of $5 \mu\text{m}$ on an aluminum plate coated with a thin layer of Vaseline (petroleum jelly) using an Eppendorf pipette."

- Lines 271-272: You mention that the two heat treatments together were used to determine the proteinaceous fraction. How? Was the heat treatment at 60°C used for anything on its own? If not, why was it done?

The first treatment at 60°C was used to estimate the bacterial fraction of the INP, while the treatment at 95°C was used to estimate the fungal fraction of the INP. We chose to solely use the proteinaceous fraction (bacterial + fungal) as this differentiation was quite scattered for a single year. Nonetheless, we have expanded the method text and added Figure S9 (Here as A7) to the SI, where the annual cycle of the bacterial, fungal and proteinaceous fraction of the INP is shown for all 4 years.

"The fraction of INP deactivated by the two heat treatments together is termed the INP proteinaceous fraction. The purpose of the two-step heat treatment was to roughly distinguish between the bacterial and fungal contribution to the total fraction of proteinaceous INPs. Bacterial INP are typically deactivated at 60°C, whereas some fungal INP withstand 60°C but are deactivated at 95°C. An annual cycle (averaged over the 4 years) for the bacterial, fungal and proteinaceous fraction is shown in

Figure S9 along with INP concentration. All four years taken together, it seems like the "fungal fraction" might be larger (52%, IQ range 39%-65%) than the "bacterial fraction" (mean 29%, IQ range 13%-47%). This analysis was based on the work by Conen *et al.*⁷⁶. We chose to work with the proteinaceous fraction only."

Figure A7: **Annual cycle of bacterial, fungal and proteinaceous fraction of INP active at $T = -12^{\circ}\text{C}$ across all four years of measurements.** Boxplots of bacterial (a), fungal (b) and proteinaceous (c, bacterial + fungal) fractions of INP active at $T = -12^{\circ}\text{C}$. Total concentration is also shown (d). For the boxplots, the continuous line represents the median and the dashed line the mean, the extent of the colored box shows the interquartile range and the whiskers the data range (1.5 times the interquartile range from the nearest quartile).

- Caption Figure S3: Panel A has a gray bar in late March – what does that bar imply?

This was a plotting bug due to values being treated as non-floats. This has been redacted and the Figure S3 updated.

- Caption Figure S4: “where only mixing-layer (ML) was considered”: How do you define the mixing-layer?

The mixing height depth is an output of the HYSPLIT model. We consider the mixing-layer (ML) as the points of the trajectory in which the altitude of the air mass is below the mixing height. A short description is given at the caption of Figure S4. The text added reads as:

"(a) 10-days back trajectories (BT) with end point at Zeppelin Observatory (ZEP), where only mixing-layer (ML) was considered. Trajectory points whose altitude sits

below the mixing height output by the model are considered to be within the ML. (b) MODIS active fire data was gridded and counted for each BT. Ensemble of trajectories pass through 12597 fires."

3 Further changes

- We converted the supporting eBC (MAAP) data using the site-specific mass absorption coefficient (MAC-value) given by Ohata et al. (2021) and accounted for the wavelength difference of the MAAP. In this manner, we are consistent with other publications from the same site. The overall results stay unchanged. We added to the revised manuscript: "The site-specific mass absorption cross section of Ohata et al. of $10.6 \text{ m}^2 \text{ g}^{-1}$ was used for the conversion from abs (Mm^{-1}) to eBC (ng m^{-3}). In addition, we accounted for the wavelength differences of the MAAP reported by Müller et al. by multiplying all eBC values with 1.05.
- Harmonization of "warm" or "high-temperature" INP. We have decided to use only the term "high-temperature INP".
- Harmonization of terms referring to correlations, favoring the nomenclature "high-low" in place of "weak-strong".
- In Figure 3, the correlations between the MBS particle groups and the biological secondary organic aerosol tracer 2-methylerythritol were wrongly performed with 2-methylthreitol. Similar in nature, 2-methylthreitol is less abundant than 2-methylerythritol. The figure has been updated to show the correlations to 2-methylerythritol. The one correlation mentioned in the text (between PBAP type II and 2-methylerythritol) is still correct and has an R value of 0.77.

4 Other updated figures

- Figure A8 (2 in the main manuscript): y-axis in panel (b) have been update for better readability.
- Figure A9 (3 in the main manuscript): correlation between MBS groups and eBC and 2-methylerythritol have been updated.
- Figure A10 (S3 in the SI): panel (a) had the grey bar removed. Panel (b) had the y axis altered to log scale and number of data points per classification added at the top.
- Figures A11-A13 (S4-S6 in the SI): are now two panel plots.
- Figure A14 (S8 in the SI): missing filter data for the month of april (when MBS was not measuring) is now shown.

Figure A8: **Multi-year trends of INP proteinaceous fraction and PBAP tracers.** a) Timeline of INP proteinaceous fraction (at $T = -12^{\circ}\text{C}$) along with a Savitzky–Golay filter curve of third order. Green background represents summer months (1st of June until the end of September) with elevated PBAP concentration as it was observed in the year 2020 by the MBS; Along with the observed air temperature (above 0°C) at Ny-Ålesund and INP concentration (at $T = -12^{\circ}\text{C}$). b) Monthly trend of INP proteinaceous fraction and concentration (at $T = -12^{\circ}\text{C}$) for four years (2017–2020) along with the PBAP concentration (2020 and some months from 2019 and 2021). c) Monthly particulate-matter concentrations of arabitol and mannitol measured over the same 4 years as panel (a) and (b). For the boxplots, the continuous line represents the median and the dashed line the mean, the extent of the colored box shows the interquartile range and the whiskers the data range (1.5 times the interquartile range from the nearest quartile).

Figure A9: Correlation coefficients between INP and, particulate matter ionic/molecular concentrations and the MBS particle groups and spectral classes. a) Spearman correlation coefficient (R) between INP concentration at temperature bins and the particles classified by the MBS. b) Spearman correlation coefficient (R) between relevant ionic/molecular concentrations in particulate matter. Solid color markers represent significant correlation ($p < 0.05$). Diamond markers represent primary biological or organic tracers. Aluminium correlations are not shown as they were not significant ($p > 0.05$).

Figure A10: **Back trajectory analysis.** (a) Daily back-trajectory contribution over land, ocean and ice superimposed by levoglucosan filter measurements. (b) Mean concentrations of several MBS measured particles categorized for more terrestrial or more oceanic (ocean+ice) back trajectories. **The numbers of points per classification are given at the top of the panel.** For the boxplots, the continuous line represents the median and the dashed line the mean, the extent of the colored box shows the interquartile range and the whiskers the data range (1.5 times the interquartile range from the nearest quartile).

Figure A11: **Back trajectory calculations cross-referenced with active fires for 7 of July of 2020.** (a) 10-days back trajectories (BT) with end point at Zeppelin Observatory (ZEP), where only mixing-layer (ML) was considered. Trajectory points whose altitude sits below the mixing height output by the model are considered to be within the ML. (b) MODIS active fire data was gridded and counted for each BT. Ensemble of trajectories pass through 12597 fires.

Figure A12: Same as Figure S4 but 26 of July of 2020. Ensemble of trajectories pass through 77091 active fires.

Figure A13: Same as Figure S4 but for 6 of October of 2020. Ensemble of trajectories pass through 42080 fires.

Figure A14: **Comparison between dust tracers and INP concentrations.** Comparison between the yearly trend through 2020 of particulate matter concentrations of aluminum (a), iron (b), magnesium (c) and calcium(d) with INP concentrations at three different ‘warm’ temperatures.

Reviewer #2 (Remarks to the Author):

Review of "Regionally sourced bioaerosols drive high-temperature ice nucleating particles in the Arctic" by Freitas et al., submitted to Nature Communications

I congratulate the authors of this manuscript. This is a very good piece of work! The issues I raised in the former review round were satisfactorily addressed and I highly recommend publication in Nature Communications.

There are only still two minor things I would like the authors to change before publication:

Line 101-102: "... and CP and FP to more oceanic regions."

While I follow your statement in the first half of this sentence, I do not see this relation for CP and FP to more oceanic regions (it was the same for the earlier, linear version of the plot). Either give numbers to prove this, or delete the part of the sentence after "and CP ..."

Line 254: Please add the type of filters used for INP measurement method II. (Quarz fiber, I assume?)

Reviewer #2 (Remarks to the Author):

Review of “Regionally sourced bioaerosols drive high-temperature ice nucleating particles in the Arctic” by Freitas et al., submitted to Nature Communications

I congratulate the authors of this manuscript. This is a very good piece of work! The issues I raised in the former review round were satisfactorily addressed and I highly recommend publication in Nature Communications.

There are only still two minor things I would like the authors to change before publication:

Line 101-102: “... and CP and FP to more oceanic regions.”

While I follow your statement in the first half of this sentence, I do not see this relation for CP and FP to more oceanic regions (it was the same for the earlier, linear version of the plot). Either give numbers to prove this, or delete the part of the sentence after "and CP ..."

The sentence “... and CP and FP to more oceanic regions.” was removed from the manuscript.

Line 254: Please add the type of filters used for INP measurement method II. (Quarz fiber, I assume?)

Yes, the filters used were Quartz fibers. This has been added and now reads as:

“The second method (Method II) is based on weekly filter (Quartz fiber) sampling downstream a PM10 inlet for the period of January 2017 to January 2021”